# Genetic Determinants of Plasma Low-Density Lipoprotein Cholesterol Levels: Monogenicity, Polygenicity, and “Missing” Heritability

**DOI:** 10.3390/biomedicines9111728

**Published:** 2021-11-19

**Authors:** Jesús Maria Martín-Campos

**Affiliations:** Stroke Pharmacogenomics and Genetics Group, Institut de Recerca de l’Hospital de la Santa Creu i Sant Pau (IR-HSCSP)–Biomedical Research Institute Sant Pau (IIB-Sant Pau), C/Sant Quintí 77-79, 08041 Barcelona, Spain; jmartinca@santpau.cat

**Keywords:** plasma cholesterol levels, autosomal dominant hypercholesterolemia, genetic risk scores, mosaicism, maternal effect, epigenetics

## Abstract

Changes in plasma low-density lipoprotein cholesterol (LDL-c) levels relate to a high risk of developing some common and complex diseases. LDL-c, as a quantitative trait, is multifactorial and depends on both genetic and environmental factors. In the pregenomic age, targeted genes were used to detect genetic factors in both hyper- and hypolipidemias, but this approach only explained extreme cases in the population distribution. Subsequently, the genetic basis of the less severe and most common dyslipidemias remained unknown. In the genomic age, performing whole-exome sequencing in families with extreme plasma LDL-c values identified some new candidate genes, but it is unlikely that such genes can explain the majority of inexplicable cases. Genome-wide association studies (GWASs) have identified several single-nucleotide variants (SNVs) associated with plasma LDL-c, introducing the idea of a polygenic origin. Polygenic risk scores (PRSs), including LDL-c-raising alleles, were developed to measure the contribution of the accumulation of small-effect variants to plasma LDL-c. This paper discusses other possibilities for unexplained dyslipidemias associated with LDL-c, such as mosaicism, maternal effect, and induced epigenetic changes. Future studies should consider gene–gene and gene–environment interactions and the development of integrated information about disease-driving networks, including phenotypes, genotypes, transcription, proteins, metabolites, and epigenetics.

## 1. Introduction

Based on evidence from epidemiological studies regarding the relationship between cholesterol and cardiovascular disease (CVD) [1], cholesterol, with nucleic acids and glucose, is one of the most cited organic molecules in the scientific literature and household conversations. A high plasma cholesterol level alone is not usually accompanied by clinical manifestations, but persistent hypercholesterolemia is strongly associated with an elevated risk of developing highly prevalent diseases like CVD—which includes myocardial infarction (MI), ischemic stroke, and peripheral vascular disease (PVD) [2]—and high blood pressure [3]. Hypercholesterolemia is negatively associated with other diseases, such as intracerebral hemorrhage [4], but the relationship between diabetes and cholesterol is controversial [5,6,7]. CVD is one of the leading causes of death in industrialized countries; for example, in Spain, where the prevalence of CVD is lower than in other Western countries [8], according to the Spanish National Statistics Institute, the leading causes of death in 2018 were ischemic heart disease in men and ischemic stroke in women [9]. Notably, hypercholesterolemia is a modifiable risk factor, so early diagnosis is crucial for decreasing cardiovascular morbimortality, and cholesterol-lowering treatments have been shown to dramatically reduce CVD risk in hypercholesterolemic subjects [10].

Cholesterol is a structural component of animal cell membranes, influencing membrane fluidity and cell signaling [11]. It is the precursor of steroid hormones, bile acids, fats, and lipophilic vitamins, but it also has a regulatory function in multiple processes, such as in the immune system, gene transcription, enzyme functions, protein degradation, signal transduction, and apoptosis [12,13]. Its multiple functions are due to its peculiar three-part structure, which has opposing hydrophilic and hydrophobic ends, and an extremely rigid four-ring central component [13].

Both a lack or excess of cholesterol can be dangerous for cells because the balance between membrane fluidity and rigidity in many cases determines important aspects of cell functions [14]. In humans, a limited ability to catabolize excess cholesterol can lead to serious health consequences, so scientific interest has focused mainly on mechanisms for regulating the uptake and synthesis of cholesterol, rather than eliminating it.

## 2. Cholesterol Metabolism

Plasma cholesterol levels depend on a balance between dietary intake/de novo synthesis and degradation/excretion [15]. There are two pathways for cholesterol input: endogenous synthesis, mainly in the liver and exogenous uptake (Figure 1).

### 2.1. The Endogenous Pathway

Endogenous cholesterol production is localized in the endoplasmic reticulum (ER) of almost all cells, although its synthesis occurs mainly in the liver, starting from acetyl coenzyme A (acetyl-CoA), continuing with the formation of 3-hydroxy-3-methylglutaryl-coenzyme A (HMG-CoA) and, in an irreversible and rate-limiting step, leading to the formation of mevalonate in a reaction catalyzed by the transmembrane enzyme HMG-CoA reductase (HMGCR) [16]. Subsequent steps are the formation of squalene, lanosterol, and cholesterol. To prevent free cholesterol accumulation, cholesterol is esterified by the enzyme acyl-coenzyme A:cholesterol acyltransferase (ACAT) [17]. In the liver, both endogenous and exogenous cholesterol can be assembled into very low-density lipoproteins (VLDLs). After the delivery of free-fatty acids (FFA) to peripheral tissues by lipoprotein lipase (LPL), VLDLs derive cholesterol-rich low-density lipoprotein (LDL) particles. Esterified cholesterol can also be transferred from high-density lipoproteins (HDLs) to LDLs by cholesteryl ester transfer protein (CETP) [18]. LDL particles are internalized in peripheral cells and hepatocytes through an endocytosis process mediated by the LDL receptor (LDLR), which recognizes the apolipoprotein (apo) B isoform apo100 of LDL and is assisted by an LDLR adaptor protein (LDLRAP1) [19]. The esterified cholesterol is hydrolyzed by enzyme lysosomal acid lipase (LAL) [20], and the resulting free cholesterol can be transferred to the membrane or the cytosol by Niemann—Pick C proteins 1 (NPC1) and 2 (NPC2) [21]. The endosome LDLR can be recycled and returned to the cell surface or degraded when the proprotein convertase subtilisin/kexin type 9 (PCSK9) complexes to LDLR [22].

### 2.2. The Exogenous Pathway

In brief, dietary cholesterol is absorbed by enterocytes in the small intestine, together with other sterols, such as plant sterols (sitosterols), via the Niemann–Pick C1-like 1 (NPC1L1) transporter [23]. Sitosterols are back-secreted to the lumen by the heterodimeric ATP-binding cassette G5/G8 transporter (ABCG5/G8) [24]. Together with triglycerides and the apoB isoform apoB48, cholesterol is packaged into chylomicrons (CMs), secreted to the lymph, and travels to the systemic circulation. The LPL enzyme hydrolyzes the triglycerides from CMs and delivers FFAs to muscle and adipose tissues, resulting in CM-remnant particles that can be internalized in hepatocytes by both LDLR and LDLR-related protein (LRP1) [25] receptors that recognize the apoE present in remnant particles. In the liver, cholesterol can be converted into bile acids and secreted back to the intestine or assembled into triglyceride-rich VLDL and resecreted into systemic circulation.

HMGCR in synthesis—by controlling the rate of cholesterol production—and PCSK9 in cellular uptake—by blocking the absorption by human cells—are critical in cholesterol homeostasis. Consequently, both proteins are main targets in cholesterol-lowering therapies, together with cholesterol receptor NPC1L1: HMGCR is the target for the lipid-lowering statin drugs, NPC1L1 for ezetimibe, and PCSK9 for protein inhibitors.

Although only 25% of the cholesterol absorbed by the intestine comes from food intake, several studies have shown a significant increase in plasma cholesterol levels associated with increased dietary cholesterol, although the existence of an exact quantitative relationship remains controversial due to the putative interaction between dietary cholesterol and other nutrients in food and the influence of gut microbiota in cholesterol adsorption [26]. Recently, a study of more than 29,000 subjects showed a significant association of dietary cholesterol and egg consumption with a higher risk of CVD and all-cause mortality in a dose-dependent manner [27].

Unlike elevated plasma cholesterol due to dietary habits, primary hypercholesterolemia involves lifelong exposure to elevated cholesterol levels while untreated, which increases the risk of CVD; therefore, the present review focuses on genetic determinants of plasma LDL cholesterol (LDL-c) levels.

## 3. Genetic Determinants of Plasma LDL Cholesterol Levels in the Pregenomic Era

The first classification of plasma lipid disorders was Fredrickson and Lee’s phenotypic classification [28]. This classification distinguished six basic phenotypes (HLP1, HLP2A, HLP2B, HLP3, HLP4, and HLP5) and was based on patterns of lipoprotein fractions observed in hyperlipidemic subjects. Notably, none of the phenotypes were characterized by alterations in HDL and, except for HLP2A, almost all phenotypes exhibited altered triglyceride-rich lipoproteins. For several years, this classification was clinically useful since different phenotypes were associated with different CVD risks [29]. The diseases associated with each phenotype were considered to be mainly caused by major-gene pathogenic variants; therefore, studies on the association between lipoprotein levels and known targeted genes were the main methodology for the study of dyslipidemias.

### 3.1. Monogenic Forms of Hypercholesterolemia

The most frequent form of plasma cholesterol alteration is hypercholesterolemia, which is defined as a persistently high plasma cholesterol level (usually ≥4.9 mmol/L). Initially, familial hypercholesterolemia (FH) corresponded to Fredrickson’s hyperlipoproteinemia type 2A phenotype (HLP2A) [28]—an autosomal dominant disorder. According to the Online Mendelian Inheritance in Man (OMIM) database, autosomal dominant hypercholesterolemia (ADH) involves phenotypes (see Table 1):FHCL1 (OMIM #143890) or defective cellular LDL receptor (LDLR) is an autosomal dominant disorder due to loss-of-function *LDLR* gene variants [30]—the most common genetic defect in FH [31,32]. Defective LDLR results in reduced LDL-c uptake by hepatocytes, with a consequent increase in blood cholesterol.FHCL2 or familial ligand-defective hypercholesterolemia (OMIM #144010) is an autosomal dominant disorder due to missense *APOB* variants [33] (mainly p.Arg3527Gln). Since each LDL particle contains only one molecule of apoB100, a ligand-defective apoB results in an inability of LDL to bind to the LDLR, impairing its clearance from the blood. Mutations of the *APOB* gene account for 6–10% of ADH cases in Europeans [32].FHCL3 (OMIM #603776) is an autosomal dominant disorder due to gain-of-function variants of LDLR catabolic regulator proprotein convertase subtilisin/kexin type 9 (PCSK9) [34]. PCSK9 is an enzyme involved in the regulation of the degradation of LDLR in the lysosome, and gain-of-function mutants increase the degradation of LDLR, reducing its quantity on the hepatocyte surface [35].There are also recessive forms of phenotype HLP2A—referred to as autosomal recessive hypercholesterolemia (ARH or FHCL4) (OMIM #603813)—mainly due to protein-truncated mutations of the low-density lipoprotein receptor adaptor-protein 1 gene (*LDLRAP1*) [19]—a cytosolic protein that interacts with the cytoplasmatic tail of the LDLR.

**Table 1 biomedicines-09-01728-t001:** Genes associated with primary monogenic dyslipidemias related to plasma LDL-c levels.

Gene	Chromosome	Phenotype ^1^	Type	Inheritance ^2^	OMIM
high LDL-c					
*LDLR*	19p13.2	FHCL1	loss-of-function	AD	#143890
*APOB*	2p24.1	FHCL2	missense	AD	#144010
*PCSK9*	1p32.3	FHCL3	gain-of-funtion	AD	#603776
*LDLRAP1*	1p36.11	FHCL4	protein-truncated	AR	#603813
Phenocopies					
*ABCG5/8*	2p21	sitosterolemia	loss-of-function	AR	#618666/#210250
*APOE*	19q13.32	FCHL/dysB	p.Leu167del		#617347
*CYP7A1*	2q35	CTX	loss-of-function	AR	#213700
*LIPA*	10q23.21	CESD/WD	loss-of-function	AR	#278000
*LPA*	6q25-q26			AD	#618807
low LDL-c					
*APOB*	2p24.1	FHBL	protein-truncated	AD	#615558
*PCSK9*	1p32.3	FHBL	loss-of-function	AD	#615558
*ANGPTL3*	1p31.3	FHBL2	loss-of-function	AR	#605019
*MTTP*	4q23	ABL	loss-of-function	AR	#200100
*SAR1B*	5q31.1	CMRD	loss-of-function	AR	#246700
Other genes					
*NPC1L1*	7p13	↓LDL-c	loss-of-function		#617966
*MYLIP*	6p22.3	↓LDL-c			*610082
*SREBF1*	17p11.2	CHL		AD	*184756

^1^ FHCL, familial hypercholesterolemia; FCHL, familial combined hyperlipidemia; dysB, dysbetalipoproteinemia; CTX, cerebrotendinous xanthomatosis; CESD, cholesteryl ester storage disease; WD, Wolman disease; FHBL, familial hypobetalipoproteinemia; ABL, abetalipoproteinemia; CMRD, chylomicron retention disease; CHL, combined hypolipidemia. ^2^ AD, autosomal dominant; AR, autosomal recessive.

Recently, a systematic analysis of the prevalence of ADH in 11 million subjects across 104 studies reported a prevalence of 1 in 313 in the general population [36]. There are no data for more than 90% of countries, but the prevalence could be higher in some countries or even within a country. In Spain, the prevalence of ADH is estimated at 1/300 [37], but data from 2.5 million primary care patients in Catalonia estimated the ADH phenotype at 1/192 [38], similar to other European countries, like the 1/217 rate reported in Denmark [39], but higher than the 1/285 estimated in another Spanish region [40]. However, estimated ADH prevalences should be compared with caution since some studies reported prevalence based on different diagnostic criteria; Zamora et al. (2017) used the plasma concentration of LDLc adjusted for age and sex as the diagnostic criteria, whereas other studies used more comprehensive clinical diagnostic scores including family history, like the Dutch Lipid Clinic Network [41] or the Simon Broome [42].

Linkage studies and/or exome sequencing in ADH-affected families have suggested other putative loci for ADH (Table 1). Linkage analysis in an ADH kindred without *LDLR*, *APOB* and *PCSK9* mutations, identified the gene for signal transducing adaptor family member 1 (*STAP1*)—a docking protein—as a candidate for ADH [43], initially describing it as FH4 (OMIM #604298). However, studies in Spanish families with a clinical diagnosis of ADH showed incomplete or lack of *STAP1* mutations cosegregation with the ADH phenotype [44,45]. A recent study failed to find *STAP1* associated with plasma LDL-c in mice or humans [46], so its exclusion from candidate genes has been proposed.

Autosomal recessive cholesteryl ester storage disease (CESD) is caused by mutations of the lysosomal acid lipase (*LIPA*) gene. One study associated a homozygous splice junction mutation of the *LIPA* gene with the ARH phenotype in a Dutch family [47], but another study of patients with a clinical diagnosis of FH detected an enrichment of heterozygous (but not homozygous) *LIPA* mutations [48,49].

Cytochrome P450 family 7 subfamily A member 1 (CYP7A1)—also known as cholesterol 7-alpha monooxygenase—is the rate-limiting enzyme that catalyzes the first step of the transformation of cholesterol into bile acids [50]. A homozygous *CYP7A1* frameshift mutation was associated with high levels of LDL-c in a UK family [51], and a promoter *CYP7A1* gene variant has been reported to influence the LDL-c-lowering response to atorvastatin, modulated by the *APOE* genotype [52].

The *APOE* mutation p.Leu167del was associated with ADH in a large French family [53]. This *APOE* variant was previously associated with splenomegaly, thrombocytopenia, and the Fredrickson HLP2B (familial combined hyperlipidemia) and HLP3 (dysbetalipoproteinemia) phenotypes [54], both characterized by mixed hyperlipidemia. A study of ADH subjects with this mutation showed that VLDLs carrying mutant *APOE* caused hypercholesterolemia by down-regulating *LDLR* expression in hepatocytes [55].

*ABCG5* and *ABCG8* loss-of-function mutations are associated with sitosterolemia (OMIM #618666/#210250)—an autosomal recessive disease characterized by elevated plant-sterol plasma levels. Although sitosterolemia shares clinical features with ADH, such as the presence of tendon xanthomas and CVD risk, cholesterol plasma levels in affected subjects are typically normal or moderately elevated in adulthood [56]. A common *ABCG5/8* polymorphism was associated with plasma lipid concentrations in ADH and influenced CVD risk [57,58]. More recent studies reported a significantly higher frequency of carriers of pathogenic or likely-pathogenic ABCG5/8 mutations in mutation-negative ADH patients compared to the reference population [59,60,61,62]. However, although ABCG5/8 mutations may contribute to hypercholesterolemia in mutation carriers, it has not been proven to be sufficient to cause an ADH phenotype [62].

Patatin-like phospholipase domain containing 5 (PNPLA5) belongs to a patatin-like phospholipase family, which plays a key role in the hydrolysis of triglycerides and the regulation of adipocyte differentiation [63]. Exome sequencing of individuals with extreme LDL-c levels showed an association of rare *PNPLA5* variants with mainly high, but also low, plasma LDL-c levels [64].

Recently, a study has described the role of cyclase-associated protein 1 (CAP1) in cholesterol metabolism [65]. CAP1 is a binding partner of PCSK9 and plays an important role in LDLR catabolism by directing LDLR–PCSK9 complex to lysosomal degradation. To my knowledge, no studies have been performed to detect *CAP1* variants associated with plasma LDL-c levels.

Finally, it is noteworthy that patients diagnosed with ADH have elevated plasma levels of lipoprotein (Lp)(a)—an LDL-like particle with apolipoprotein(a) covalently bonding to apoB [66]. In the absence of hypertriglyceridemia, plasma LDL-c concentrations are usually calculated using the Friedewald formula [67], rather than by direct detection; thus, Lp(a) particles could be responsible for an increased likelihood of high LDL-c diagnosis, since the cholesterol within Lp(a) contributes to the estimated LDL-c. A recent study showed that the presence of Lp(a) cholesterol misclassified a significant number of samples submitted for lipid testing as high LDL-c [68]. Lp(a) is an independent CVD risk factor, and 90% of circulating Lp(a) plasma levels are genetically determined [69], so studies must consider a possible interference of Lp(a) if plasma LDL-c concentrations are not directly measured.

### 3.2. Monogenic Forms of Hypocholesterolemia

A very low plasma cholesterol level, or hypocholesterolemia, is usually defined as persistent plasma total cholesterol, LDLc, and apoB concentrations below the 5th percentile of the reference population [70]. Hypocholesterolemia is a rare condition and, in the heterozygous form, usually does not have a clinical expression, so it is usually underdiagnosed. The main disorders related to low plasma cholesterol levels are as follows (Table 1):FHBL or familial hypobetalipoproteinemia (OMIM #615558) is an autosomal codominant disorder caused mainly by protein-truncated *APOB* gene mutations [71], but also by loss-of-function *PCSK9* gene mutations [72];FHBL2 or familial hypobetalipoproteinemia type 2 (OMIM #605019)—also known as familial combined hypolipidemia—is an autosomal recessive disorder caused by loss-of-function mutations of the angiopoietin-like 3 (*ANGPTL3*) gene [73]. ANGPTL3 is an inhibitor of the lipases LPL and LIPG (endothelial lipase), reducing the clearance of triglyceride-rich particles [74];ABL or abetalipoproteinemia (OMIM #200100) is an autosomal recessive disorder caused by mutations of the microsomal triglyceride transfer protein (*MTTP*) gene [75]. MTTP is a chaperone and the major lipid transfer protein of triglyceride, cholesteryl esters, and phospholipid to nascent apoB-containing lipoproteins [76];Chylomicron retention disease (CMRD; OMIM #246700)—also known as Anderson’s disease—is an autosomal recessive disorder caused by mutations of the secretion-associated Ras-related GTPase 1B (*SAR1B*) gene [77]. SAR1B plays a central role in the specific prechylomicron transport vesicles within the Golgi apparatus, as a component of coat protein complex II [78].

Other candidate loci for hypocholesterolemia have been proposed. Apolipoprotein C-III (apoC-III) reduces the clearance of triglyceride-rich particles by inhibiting lipoprotein (LPL) and hepatic (LIPC) lipases [79]. Loss-of-function *APOC3* variants have been associated with low LDL-c and reduced CVD risk [80,81], but the results of a multi-ethnic study failed to detect any association with low LDL-c [82]. The reduction in CVD risk seems to be related to reduced remnant cholesterol, rather than low LDL-c [83].

Myosin regulatory light chain interacting protein (MYLIP)—an E3 ubiquitin ligase—promotes the degradation of LDLR. A loss-of-function *MYLYP* mutation was associated with low plasma LDL-c in a Dutch cohort [84]; however, a *MYLIP* common missense mutation (p.Asn342Ser) was recently associated with hypercholesterolemia in Han Chinese people [85].

As we mentioned previously, enterocytes in the intestine incorporate dietary sterols via the NPC1L1 transporter. Targeted sequencing of *NPC1L1* identified loss-of-function mutations associated with low cholesterol absorption, low plasma LDL-c, and a lower risk of CVD [86,87]. Notably, the NPC1L1 transporter is the target for the lipid-lowering drug ezetimibe.

Sterol regulatory element-binding transcription factor 1 gene (*SREBF1*) codifies SREBP1—a transcription factor that regulates cholesterol and fatty acid synthesis in the liver. An *SREBF1* missense mutation (p.Pro111Leu) was identified in a family with severe combined hypolipidemia [88], alongside reduced transcriptional activation of *LDLR*, *ABCA1*, fatty acid synthase (*FAS*), *MTTP*, and *HMGCR*.

Rare mutations in these genes may explain extreme cases at both ends of the plasma LDL-c distribution, but not the less severe and more common LDL-c-related dyslipidemias. The inheritance of moderate conditions does not follow a clear Mendelian pattern; thus, the next section aims to identify the genetic basis of the population variance in LDL-c.

## 4. Genetics of LDL Cholesterol in the Big-Data Era

In the late twentieth century, the development of high-throughput genotyping platforms, such as DNA microarrays, made it possible to analyze many variants simultaneously. Additionally, the publication in 2003 of the human genome sequence and the development of next-generation sequencing (NGS) have made it affordable to sequence the genomes of a growing number of individuals from different ethnic groups and to identify genetic variability in human populations. Jointly with the development of bioinformatics tools and statistical methods to analyze large numbers of samples, these advances have led to the possibility of performing genome-wide association studies (GWASs), making it possible to detect which regions of genomes explain variances in quantitative traits or are risk factors for common diseases (Figure 2). GWASs, in most cases, replace previous family linkage studies that used genomic markers (generally microsatellites) to detect candidate regions. Linkage studies look for chromosomal segments that cosegregate with a given trait, using extended pedigrees and analyzing several generations of affected families, assuming few recombination events and low recurrent mutation rates. The resulting regions used to be extensive, but this method is still suitable for highly penetrant phenotypes, typical of most rare diseases. However, GWASs look for traits associated with a given chromosomal segment, using unrelated individuals and, thus, considering many generations of recombination. Unlike microsatellites, which have higher mutation rates, most variants used in GWASs are single-nucleotide variants (SNVs) that can be considered identical by descent. This works well for low penetrant phenotypes, typical of most common diseases [89].

The basis of candidate gene-targeted analysis is the hypothesis that such genes are involved in a trait. By contrast, GWASs are hypothesis generators that do not depend on previous knowledge. The variants used in association analysis must pass quality controls. In general, SNVs presenting a minor allele frequency (MAF) <1%, Hardy–Weinberg p-values <1 × 10^−4^ in the control/general population group, and missing genotypes >5% are excluded from the analysis [90]. Associations are considered highly suggestive if the p-value is ≤1 × 10^−5^, and significant if it is ≤5 × 10^−8^, based on studies of European populations using HapMap data [91]. This threshold depends on the number of common variants used (defined as a MAF ≥ 0.05) and the level of linkage disequilibrium (LD), so a more stringent threshold is recommended when including less frequent variants or studying populations with lower LD between common variants. The imputation strategy [92]—an in silico method to infer missing genotypes using haplotype information from a reference panel—allows the number of genomic variants for association testing to be increased. Candidate genes are considered within 1 Mb around each significant association. Nowadays, it is mandatory to replicate significant regions in an independent sample.

GWAS summary statistics can be downloaded from the GWAS Catalog [93] or the GWAS Central [94] website. In the GWAS Catalog (www.ebi.ac.uk/gwas, accessed on 18 August 2021), 139 studies and 3915 associations were found regarding the trait “low-density lipoprotein cholesterol measurement” (EFO_0004611). Filtering for reported trait and discovery samples (see Appendix A for details) resulted in 22 studies and 696 associations with a *p*-value of ≤1 × 10^−5^ (at least highly suggestive) for Europeans or European ancestry populations (Appendix A), 10 studies and 131 significant associations for East Asians (Appendix A), three studies and 21 associations for Afro-Americans or African ancestry (Appendix A), two studies and 14 associations for Native Americans or Hispanics (Appendix A), one study and seven associations for the Middle-Easterners (Appendix A), and two studies and four significant associations for Oceanians (Appendix A). The supplementary tables only include variants from the GWAS catalog and exclude multiethnic studies. For more information about a specific study, the summary statistics can be downloaded if authors have made them available.

### 4.1. Genes Related to LDL Cholesterol Metabolism

Some “classical genes” used in previous targeted analyses, such as *LDLR* and *LDLRAP1* for hypercholesterolemia, *ANGPTL3* for hypocholesterolemia, and *APOB* and *PCSK9* for both hypo- and hypercholesterolemia, have shown significant associations, while others, such as *MTTP* and *SAR1B*, have not. Since GWASs usually only consider common variants, a possible explanation is that some genes only present rare variants associated with LDL-c, such as those identified in whole-exome sequencing studies [64]. Only a small proportion of the SNVs associated with the trait in GWASs (usually less than 10%) map to the coding region of a gene, and most of them are annotated as non-pathogenic variants. Since most of the SNVs included in GWASs do not seem to alter the function or the expression of genes, it is assumed that they are in LD with undetected functional variants, not ruling out that some SNVs located outside coding regions may affect regulatory sequences of the expression of nearby genes. However, it is notable that some measures of plasma LDL-c levels used in GWASs are obtained from non-fasting subjects, as in the case of studies using samples from the UK biobank (UKBB).

**Figure 2 biomedicines-09-01728-f002:**
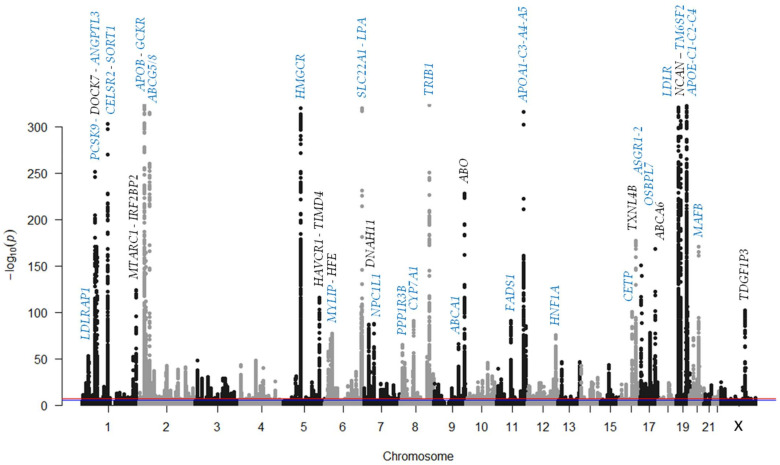
An example of a genome-wide association study (GWAS). A Manhattan plot was constructed from data taken from the summary statistics of Klimentidis et al. [95]—a GWAS performed on samples from the UK biobank. Only loci with *p*-values ≥ 1 × 10^−50^ are labeled. Genes related to lipid metabolism are labeled in blue. The red line marks the significant threshold (*p*-value ≤ 5 × 10^−8^) and the blue line marks the highly suggestive threshold (*p*-value ≤ 1 × 10^−5^).

#### 4.1.1. Apolipoproteins and Lipoprotein Receptors

Significant associations were obtained for some apolipoprotein genes. Soluble apolipoproteins are structurally similar and, except for apoA-II, their genes are distributed in two clusters in chromosomes 11q23.3 (genes for apoA-I, C-III, A-IV, A-V) and 19q13.31-32 (apoE, C-I, C-II, C-IV) [96]. Both clusters are associated with plasma LDL-c levels. ApoA-I and apoA-IV are implicated in cholesterol reverse transport, and apoA-V is an important regulator of plasma triglyceride levels. Apo-CIII counteracts the effects of apoC-II and inhibits clearance of triglyceride-rich lipoproteins (TRLs). *APOC2* is located in the other cluster, and its product is necessary to activate the LPL and subsequent clearance of TRLs. ApoC-I inhibits lipoprotein uptake via LDLR receptors, while the function of ApoC-IV is unclear. ApoE is a multifunctional apolipoprotein, and regarding cholesterol metabolism, regulates cholesterol homeostasis, acting as a ligand for LDLR, VLDLR, and TRL remnant receptors. The *APOE* variant rs7412, which determines the ε2—an apoE isoform with less affinity to receptors than ε3 and ε4—is a strong significant association common to Europeans, East Asians, and Afro-Americans (see ). Another apolipoprotein significantly associated with LDL-c is apoH (*APOH*, Chr 17), which is poorly associated with plasma lipoproteins, although it was recently described as interacting with PCSK9 [97]. Finally, the apo(a) gene (*LPA*, Chr 6) is also associated with plasma LDL-c levels. As mentioned above, apo(a) is a component of LDL-like Lp(a) particle and an independent risk factor for CVD.

Association with the genes for lipoprotein receptors LDLR (Chr 19), LDLRAP1 (Chr 1), LDL receptor-related protein associated protein 1 (LRPAP1, Chr 4), and very low-density lipoprotein receptor (VLDLR, Chr 9) were also obtained. LRPAP1 has an inhibitory effect on ligand binding to members of the LDLR family LRPA1 and LRPA2 by CM remnants [98]. VLDLR binds apoE containing triglyceride-rich lipoproteins VLDL and intermediate-density lipoprotein (IDL).

#### 4.1.2. Transporters

ATP-binding cassette subfamily A member 1 (ABCA1, Chr 9) is a member of the ATP-binding cassette (ABC) transporter family, with 48 transcriptionally active genes divided into seven subfamilies, and with an important function in the regulation of the transport of diverse molecules across membranes. ABCA1 plays an important role in cellular cholesterol removal, and *ABCA1* gene mutations cause Tangier disease (OMIM #205400)—an autosomal recessive disease characterized by reduced levels of HDL-c and intracellular accumulation of cholesteryl esters [99]. The downregulation of *ABCA1* is associated with glomerular accumulation of cholesterol in diabetic kidney disease [100]. ABCA6 (chr 17) is another ATP-binding cassette responsive to cholesterol, and its putative role in intracellular lipid transport have been proposed [101], although a recent study showed no role of Abca6 in lipoprotein metabolism in rodents [102], so its implication in cholesterol metabolism is still unclear. ABCG5 and ABCG8 (chr 2) are other members of the ABC family. As mentioned in Section 2.2, in a heterodimeric form, they have an important function in the back-secretion of plant sterols to the small intestine lumen. Other transporter genes significantly associated with LDL-c are *NPC1L1* (Chr 7) and *NPC1* (Chr 18), which participate in exogenous and endogenous pathways, respectively (see Section 2). Finally, the genes for two plasmatic lipid transfer proteins—the cholesteryl ester transfer protein (CETP, Chr 16), and the phospholipid transfer protein (PLTP, Chr 20)—are also associated with LDL-c. CETP transfers cholesteryl esters from HDLs to LDLs, and PLTP transfers phospholipids from TRLs to HDLs, contributing to the transformation of VLDL into LDL.

#### 4.1.3. Enzymes

The above-mentioned enzymes—HMGCR (Chr 5), LPL (Chr 8), and CYP7A1 (Chr 8)—are associated with plasma LDL-c levels in GWAS. Significant signals for the other members of the triglyceride lipase family, such as hepatic lipase (LIPC, Chr 15) and endothelial lipase (LIPG, Chr 18), have also been obtained. Fatty acid desaturases (FADS) are involved in the synthesis of highly unsaturated fatty acids, contributing to membrane phospholipid biosynthesis. Three members of the FADS group (FADS1, 2 and 3, Chr 11) are located in a cluster (11q13.2) and several SNVs in this region are associated with LDL-c.

#### 4.1.4. Transcription Factors and Transcriptional Modulators

Some transcription factors regulating the expression of genes involved in lipid metabolism produced significant signals in GWASs, such as the Insulin-induced gene 2 (*INSIG2*, Chr 2), which regulates sterol regulatory element-binding factors (SREBPs) processing during nutrient starvation in the livers of mice [103]. SREBPs are transcriptional regulators of many enzymes, receptors, and other proteins involved in lipid metabolism. The HNF1 homeobox A or hepatic nuclear factor 1-alpha (HNF1A, Chr 12) is a transcription factor that regulates various liver enzymes, such as HMGCR; CYP7A1; apolipoproteins B, A-I, A-II, and C-III genes; and LIPC [104], and plays a prominent role in *PCSK9* transcription [105]. HNF1A mutations cause maturity-onset diabetes of the young (MODY) type 3 (MODY3, OMIM #600496). The hepatocyte nuclear factor 4-alpha (HNF4A, Chr 20) is a regulator of CYP7A1 gene transcription among other genes [106]. HNF4A mutations cause MODY type 1 (MODY1, OMIM #125850). The tribbles pseudokinase 1 (TRIB1, Chr 8) is a transcription factor involved in macrophage lipid accumulation by promoting oxidized LDL uptake [107].

Peroxisome proliferator-activated receptors (PPARs) alpha (PPARα, Chr 22) and gamma (PPARγ, Chr 3) are ligand-activated transcription factors of the nuclear receptor superfamily. PPARs are activated by unsaturated fatty acids and form heterodimers with retinoid X receptors (RXRs) to regulate the peroxisomal beta-oxidation of fatty acids [108]. PPARα promotes fatty acid oxidation, and PPARγ increases lipogenesis and adipogenesis. Retinoic X receptor beta (RXRβ, Chr 6) is another nuclear receptor. RXRβ forms heterodimers with retinoic acid receptors (RARs) and other nuclear receptors, modulates the metabolism of lipids and bile acids in the liver, and regulates cholesterol transport in macrophages [109]. MAF bZIP transcription factor B (MAFβ, Chr 20) promotes macrophage cholesterol efflux by upregulating ABCA1 [110].

#### 4.1.5. Other Genes Related to LDL-c Metabolism

Other genes with a well-known role in cholesterol homeostasis are associated with LDL-c in GWAS, such as two proteins participating in LDLR degradation—PCSK9 (Chr 1) and MYLIP (Chr 6)—and the inhibitor of lipase activity ANGPTL3 (Chr 1). Asialoglycoprotein receptor 1 (ASGR1, Chr 17) mediates the endocytosis and lysosomal degradation of plasma glycoproteins in hepatocytes. Recently, ASGT1 was identified as a PCSK9-independent ligand of LDLR [111]. The region of cadherin EGF LAG seven-pass G type receptor 2 (*CELSR2*, Chr 1) and Sortilin 1 (*SORT1*) genes presents one of the strongest and reproducible associations with LDL-c. The *CELSR2* variant rs12740374 (c.*919G>T) is a common GWAS significant signal in European, East Asian, African, and Hispanic populations. SORT1 plays a significant role in lipid uptake, and it seems to be involved in the regulation of hepatic secretion of VLDL [112]. Although the significant association in the *CELSR2*-*SORT1* region is usually attributed to *SORT1*, a study claimed that CELSR2 deficiency suppresses lipid accumulation in hepatocytes [113]. The region of the neurocan (*NCAN*, Chr 19) gene includes the gene for transmembrane 6 superfamily member 2 (*TM6SF2*). Neurocan is a brain chondroitin sulfate proteoglycan involved in cell adhesion and migration. ApoE is the ligand of TRL remnants for heparan sulfate proteoglycans [114], but to my knowledge, no connection has been established between NCAN and apoE or lipid metabolism. On the contrary, TM6SF2 plays role in the regulation of hepatic lipid content [115,116]. Oxysterol binding protein-like 7 (OSBPL7, Chr 17) is a member of the oxysterol-binding protein (OSBP) family—a group of lipid-binding proteins involved in intracellular lipid transfer. OSBPL7 is also a regulator of the ABCA1 transporter and a putative target for drugs that restore ABCA1-dependent cholesterol efflux in kidneys [117]. The solute carrier family 22 member 1 (*SLC22A1*, Chr 6) gene codifies the organic cation transporter 1 (OCT1)—a translocase across the plasma membrane of hepatocytes of organic cations like N-1-methylnicotinamide (MNM), among others, and drugs like metformin. The signal in the *SLC22A1* region is frequently attributed to the nearby *LPA* gene, but the *SLC22A1* gene is transactivated by HNF4α [118] and its expression is regulated by RAR/RXR heterodimers [109], which also makes it a candidate for association with plasma LDL-c levels. Using network density analysis, Baillie et al. also detected strong coexpression of *SLC22A1* promoter and elements associated with LDL-c [119].

Glucose and lipids metabolisms are closely linked. It is well known that patients with ADH have a lower risk of developing type 2 diabetes than normolipidemic subjects [120]. The underlying mechanisms of such a link are not completely understood, but there is an established correlation between liver fat and insulin resistance [6]; thus, it is not surprising that some genes related to glucose metabolism are associated with LDL-c. The glucokinase regulatory protein (GCKR, Chr 2) regulates the enzyme glucokinase—an enzyme acting as a glucose sensor. During fasting, GCKR inhibits GCK activity by forming an inactive complex that dissolves after a meal, releasing GCK to the cytoplasm. Another significant association is located at the locus for the protein phosphatase 1 regulatory subunit 3B (PPP1R3B, Chr 8), which is involved in glycogen synthesis.

### 4.2. Genes Not Related to LDL Cholesterol Metabolism

Some significant signals in GWASs mapped to regions without a gene directly related to cholesterol metabolism. ST3 beta-galactoside alpha-2,3-sialyltransferase 4 (ST3GAL4, Chr 11) plays a key role in the synthesis of E-selectin ligands, influencing enzyme liver concentrations [121]. The ABO blood group (*ABO*, Chr 9) gene is also associated with serum soluble E-selectin levels [122], and there is strong evidence linking ABO and CVD risk, including in FH patients [123]. However, the exact causal mechanisms of such associations are still unknown. Other genes are: serpin family A member 1 (*SERPINA1*, Chr 14) encoding alpha-1 antitrypsin; NYN domain and retroviral integrase containing (*NYNRIN*, Chr 14); mitochondrial amidoxime reducing component 1 (*MTARC1*, Chr 1); interferon regulatory factor 2 binding protein 2 (*IRF2BP2*, Chr 1); T-cell immunoglobulin and mucin domain-containing 4 (*TIMD4*, Chr 5)–hepatitis A virus cellular receptor 1 (*HAVCR1*) region; hemochromatosis (*HFE*, Chr 6); major histocompatibility complex (*HLA*, Chr 6); dynein, axonemal, heavy chain 11 (*DNAH11*, Chr 7); thioredoxin-like 4B (*TXNL4B*, Chr 16)–haptoglobin-related protein (*HPR*) region; and teratocarcinoma-derived grow factor 1 pseudogene 3 (*TDGF1P3*). The roles of these genes in lipid metabolism have not been identified and may reflect unknown mechanisms in lipid homeostasis, so further studies regarding these loci are needed.

Despite the strong association between LDL-c and common genetic variants, GWASs have shown only a small phenotypic effect of such variants. We cannot rule out the possible existence of a new gene with a large effect on plasma LDL-c, but it is unlikely that it could explain the many cases without detected mutations of know genes; thus, another possible explanation was posited for such cases.

### 4.3. Polygenicity and Polygenic Risk Scores

Quantitative traits are complex multifactorial traits that depend on both genetic and environmental factors. In some cases, genetic factors consist of a major gene with a large effect, acting jointly with some genes modulating its effect (modifiers), but they usually consist of a set of genes with different low to moderate effects. In general, common variants identified in GWASs presented a small effect size on LDL-c, but a burden of genetic LDL-c-raising variants has been shown to exert a cumulative influence on plasma LDL-c determination [29,124]. Using an additive model, the accumulation of susceptibility variants could be measured by polygenic risk scores (PRSs), which could be either weighted—considering a different LDL-c-raising effect for each allele—or unweighted—simply counting the number of susceptibility alleles at each locus. Regarding the number of SNVs, PRSs could include the whole genome to only a few SNVs, since there are no established rules. The larger the number of SNVs, the larger the genomic information the score refers to, so the question arises as to what size a PRS should be to minimize the loss of information regarding the overall polygenic association in the population. A PRS using 10 LDL-c-raising alleles explained 3.4% of the variance in LDL-c in three European cohorts [125]. Another PRS with 37 SNVs explained 6.0% of the variation in plasma LDL-c levels in the Atherosclerosis Risk in Communities (ARIC) cohort, comprising white and black people from four US communities [126], whereas a PRS including 3.6 million variants explained 3.5% of the LDL-c variability in a Japanese ADH cohort [127]. Regarding the use of PRSs in clinical practice, an important question that has not yet been resolved is the definition of what constitutes a “high” score—the top quartile, the top decile, or another calculated cutoff point. As derived from the GWAS Catalog, there is a bias toward European populations in using SNVs associated with LDL-c to construct PRSs, which can make it difficult to apply them to populations of different ethnic origins. In a study using European, Asian, Japanese, and Ugandan cohorts, the range of reproducibility rates for LDL-c associated loci was 61.5%–77% [128]. Differences in LD structure, frequency of SNVs, and gene–environmental interactions could account for such differences.

Two main small-scale scores were used in the literature (Table 2)—a PRS proposed by Talmud et al. [129] using effect sizes from the Global Lipid Genetics Consortium (GLGC), published in Teslovich et al. [130], and a PRS proposed by Wang et al. [131], using effect sizes from Kathiresan et al. [132]. Both scores are relatively easy to set up in most clinical laboratories and have been used to assess the proportion of clinically diagnosed ADH patients with mutation-negative genetic diagnoses that may have a polygenic origin. However, the high overlap of PRSs between ADH patients and the general population suggests that the intervention of other factors—possibly environmental—is necessary to develop the hypercholesterolemic endophenotype, and makes it difficult to translate a PRS into an individual diagnosis.

Studies of ADH patients without *LDL*, *APOB*, or *PCSK9* pathogenic mutations showed that 20%–80% presented high PRSs relative to a control population [60,127,129,131,133,134,135,136,137]. Although each variant had a small effect, the accumulation of LDL-c-raising variants could lead to a phenotype similar to that of monogenic ADH [138]. By contrast, some authors pointed out that, despite the higher PRSs in ADH mutation-negative patients than in controls and ADH mutation-positive patients, the generally small percentage of explained variation in plasma LDL-c levels did not support polygenicity as a cause of the ADH phenotype [136]. LDL-c PRSs are also associated with CVD risk in mutation-negative ADH patients, who have a lower CVD risk than patients with a monogenic forms, so PRSs could be useful for establishing CVD risk stratification for ADH [139].

## 5. Other Causes of High Plasma LDL Cholesterol Levels

### 5.1. Somatic Mutations/Mosaicism

It is assumed that uncorrected copy errors can occur during DNA replication, which are called mutations. The number of new mutations that can be generated therefore depends on the number of cell divisions and may be more frequent during stages of greater meiotic/mitotic activity, such as gametogenesis, embryonic development, childhood, or adolescence, but also during adulthood. When new genetic variants appear post-zygotically, two or more cellular populations with differences in their genomes can coexist in an individual, which is called “genetic mosaicism.” This phenomenon must be differentiated from chimerism—the heterogeneity that can occur when an individual derives from two or more fertilization events. The degree of alteration in the phenotypes of such mutations depends not only on affected genes but also on the developmental timing of the mutation, the cell lineages affected, and the relative percentage of mutated cells. The earlier in embryonic development a mutation occurs, the more cell numbers and types will be affected. The variant must be present in the appropriate cell type, and the number of mutation-carrier cells must also achieve a sufficient proportion to be clinically relevant. Usually, these cases only affect individuals in whom the mutations occur, but in some cases, a mutation can affect the germline and pass to the offspring. The role of post-zygotic somatic mutations in tumor development has been widely studied, but they could also be involved in the development of many other diseases, although they are often ignored as a source of variation [140].

Mosaicism could be another genetic cause of altered plasma cholesterol levels in cases of negative results from germline genetic analyses. The liver is the main target tissue for the analysis of alterations in cholesterol metabolism, but a liver biopsy is highly invasive and is not justified for this type of analysis. An alternative method is the bioinformatic analysis of data from the NGS of blood or saliva samples, which allows the detection of somatic mutations that escape classical Sanger sequencing [141]. Recently, a somatic pathogenic *LDLR* variant was detected by NGS analysis in a 57-year-old patient with a clinical suspicion of ADH [142]. In this case, the variant affected the germline and was present in the offspring.

Further studies must establish the impact of mosaicism in hypercholesterolemia. Including the detection of mosaicism in genetic studies or reanalyzing NGS results in patients who are negative for known gene variants, especially in those cases with doubtful family history, could give an idea of the extent of post-zygotic variants in plasma cholesterol level alterations.

### 5.2. Maternal Effect

It is well known that early-life environmental exposures influence disease risk in adults. During pregnancy, a normal physiological increase in maternal cholesterol levels meets the high cholesterol demands of the fetus [143,144]. This increase is higher in women with existing hypercholesterolemia before conception, so we refer to ‘maternal hypercholesterolemia’ as a plasma cholesterol level exceeding that observed in healthy women.

In humans, maternal hypocholesterolemia is associated with preterm delivery and low weight [145], which are associated with an increased risk of cardiometabolic disease in adulthood [146,147], and maternal hypercholesterolemia is also associated with preterm delivery [148] and with a higher risk of CVD. Atherosclerotic lesions begin early in life, and maternal hypercholesterolemia may potentiate fatty-streak formation and its rate of progression [149]. Maternal inheritance of the pathogenic *LDLR* variant p.Val408Met has been associated with a significantly higher standardized mortality rate than paternally inherited ADH [150], supporting the hypothesis of a fetal origin for atherosclerosis. Indeed, some studies have reported that CVD risk depends on the sex of the hypercholesterolemic parent. Surprisingly, a recent study suggested that, in genetically confirmed ADH kindreds, paternal inheritance of the ADH-causing mutation (*n* = 321) decreases the time onset of a cardiovascular event compared with maternal inheritance (*n* = 268) [151]. However, not all studies have found this association. A study including 1231 heterozygous ADH-mother offspring showed that severe maternal hypercholesterolemia during pregnancy was not associated with a worse cardiovascular phenotype in the offspring [152].

Moreover, although the influence of maternal hypercholesterolemia on the plasma cholesterol levels of newborns is not well established [153,154], maternal hypercholesterolemia in mice has been associated with dyslipidemia in offspring, independently of obesity or a high-fat diet [155].

### 5.3. Epigenetic Modifications

Epigenetic changes are a possible mechanism of lipid gene regulation [156,157,158]. The attachment of a methyl group to a cytosine-guanidine dinucleotide (CpG) of DNA (DNA methylation) and changes in histone proteins are the best known epigenetic modifications, usually caused by environmental factors. As a result, a promoter may be modified or regions of DNA may not be exposed to the transcription machinery, resulting in alterations in gene expression. Unlike inherited genetic mutations, epigenetic modifications are dynamic and can be reversed. However, methylation changes can cause dyslipidemia but also can be a consequence of elevated lipid levels [159]; for example, an exposure of monocytes to oxidized LDL induces a proatherogenic macrophage phenotype via epigenetic histone modifications [160]. Gene variants can also alter local histone marks, affecting chromatin activity and gene expression [161].

CpG methylation seems to regulate the expression of some of the genes involved in LDL-c metabolism, such as *APOE* [162] and *NPC1L1* [163]. However, several epigenome studies showed that neither *APOE* nor *NPC1L1* were associated with LDL-c plasma levels. As in genotyping, the degree of methylation can be analyzed in targeted genes or the entire genome, allowing epigenome-wide association studies (EWASs). Hedman et al. found methylation in 23 CpGs associated with LDL-c in EWASs, located in the gene for the rate-limiting enzyme in steroid biosynthesis squalene monooxygenase (SQLE) [157]. A recent EWAS showed a significant positive association between apoB lipoproteins and methylation levels at CpG sites located in the transcription factor *SREBF1* gene and a negative association between apoB lipoproteins and CpG sites in the gene for carnitine palmitoyltransferase 1A (*CPT1A*) [164].

Differential DNA methylation could contribute to the hypercholesterolemic endophenotype in ADH patients without *LDLR*, *APOB*, and *PCSK9* gene mutations. A recent study observed no differences in methylation genes between mutation-negative and mutation-positive ADH patients, except for the carnitine palmitoyltransferase 1A (*CPT1A*) gene [165]. The authors also showed that the accumulation of genome-wide methylation differences, rather than differences in target genes or specific CpG sites, was associated with the ADH phenotype in mutation-negative patients. A high overlap between lipid-associated variants and methylation quantitative trait loci, especially in the liver [159], points to the hypothesis that the excess of LDL-c-raising variants observed in ADH mutation-negative patients can exert its effect through epigenetic modifications. Further studies are needed to elucidate whether the observed genome-wide differences in methylation and frequency of common variants are part of the same process or contribute independently to the hypercholesterolemic endophenotype.

## 6. The Future: The Era of System Genetics

GWAS studies have uncovered a large number of genes associated with plasma levels of LDL-c. Some of these genes encode proteins involved in well-known metabolic pathways related to cholesterol metabolism. However, other genes encode proteins involved in metabolic pathways whose connection to cholesterol metabolism is unknown. Thus, high-performance techniques, such as hypothesis generators, have suggested new metabolic pathways connected with cholesterol metabolism. Further functional studies are needed to determine if the identified associations are causal since identifying new therapeutic targets depends on this relationship. On the other hand, there is a bias in the published studies towards European populations. Performing GWASs in different ethnic groups is necessary to determine to what extent the genetic determinants of plasma LDL-c levels are population-specific or can be generalized.

Plasma LDL-c levels, as a complex trait, depend on both genetic and environmental factors. Usually, genetic and environmental influences on a quantitative trait are measured independently. As shown by this review, the percentage of LDL-c-related dyslipidemias that can be explained by genetic causes has been increasing over time, mainly due to the continuous development of new analytical methods. However, a significant proportion of the population variance in plasma LDL-c and many other complex traits remains unexplained by genetic inheritance and environmental effects, which has led to the concept of “missing” heritability. Gene–environmental interactions are not usually considered in association analysis, despite the evidence that the effect on the phenotype of some common variants markedly depends on the environment: for example, the *APOE* genotype influences the lowering of LDL-c in response to dietary changes in fatty acids consumption [166], and common SNVs in *ABCG5/G8* modulate plasma lipid concentrations that depend on the smoking status of ADH patients [58]. Another study showed some evidence of an interaction between a weighted PRS constructed with 32 SNVs associated with LDL-c and diet quality in a Swedish cohort [167]. Gene-gene interactions must also be considered.

Unraveling the nature of missing heritability requires new and revolutionary approaches to defining disease-driving molecular processes [168]. The use of genome-wide network studies [89], defined as the integration of information about phenotypes, genotypes, transcription, proteins, metabolites, and epigenetics, is expected to help in discovering the interconnected factors in disease-driving networks. Of course, studies should not forget the role of evolution, since the human genome is fundamentally a historical record that must be read [169].

## Figures and Tables

**Figure 1 biomedicines-09-01728-f001:**
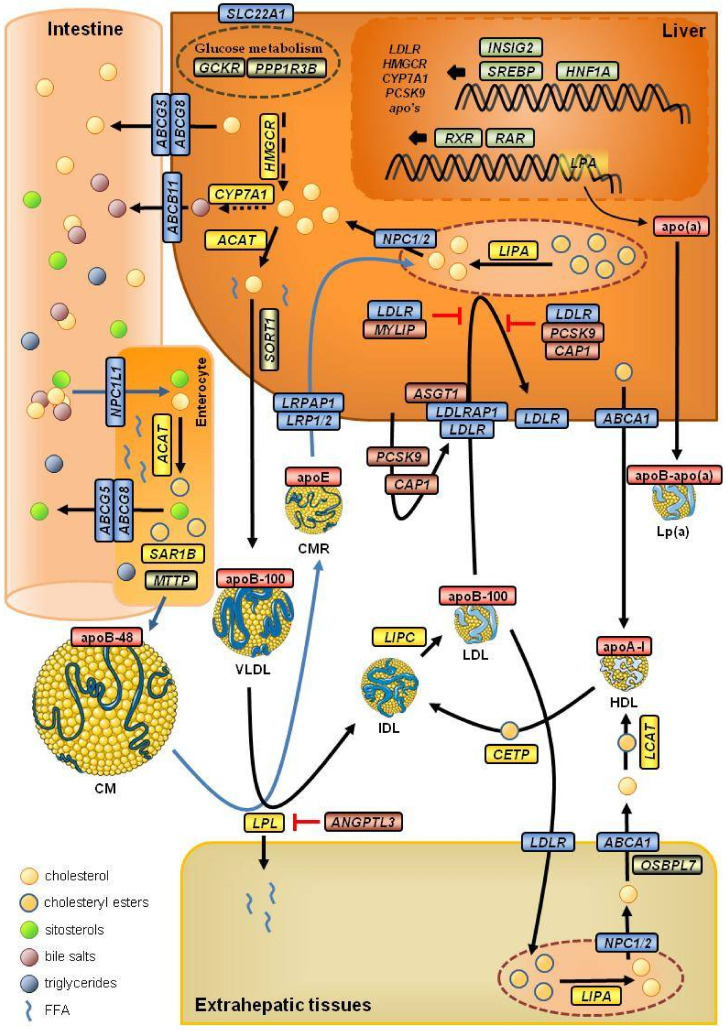
Role of the main genes associated with plasma levels of low-density lipoprotein cholesterol. Blue arrows indicate the cholesterol exogenous pathway. CM, chylomicron; CMR, chylomicron remnant; FFA, free-fatty acids; HDL, high-density lipoprotein; IDL, intermediate-density lipoprotein; LDL, low-density lipoprotein; VLDL, very low-density lipoprotein. Color codes: red, Apolipoproteins; blue, genes encoding receptors, transporters, or their associated proteins; yellow, genes encoding proteins with an enzymatic function; green, genes encoding transcription factors and transcription modulators; brown or grey, genes encoding regulatory proteins.

**Table 2 biomedicines-09-01728-t002:** SNVs associated with LDL-c plasma level used in two gene score calculations.

Gene	ID	Position ^1^	Type	Change ^2^	MA ^3^	Risk Allele	Effect ^4^
PRS from Talmud et al., 2013 [129]							
*PCSK9*	rs2479409	1:55,038,977	5’-UTR	c.-861G>A	G	G	0.052
*CELSR2*	rs629301	1:109,275,684	3-’UTR	c.*1635G>T	G	T	0.146
*APOB*	rs1367117	2:21,041,028	missense	c.293C>T (p.Thr98Ile)	T	T	0.105
*ABCG8*	rs4299376	2:43,845,437	intronic	c.166-718G>T	G	G	0.071
*HFE*	rs1800562	6:26,092,913	missense	c.845G>A (p.Cys282Tyr)	A	G	0.057
*MYLIP*	rs3757354	6:16,127,176	upstream	c.-2147C>T	T	C	0.037
*SLC22A1*	rs1564348	6:160,157,828	intronic	c.1599-688T>C	C	T	0.050
*ST3GAL4*	rs11220462	11:126,374,057	intronic	c.-61+18215G>A	A	A	0.050
*NYNRIN*	rs8017377	14:24,414,681	missense	c.2932G>A (p.Ala978Thr)	A	A	0.029
*LDLR*	rs6511720	19:11,091,630	intronic	c.67+2015G>T	T	G	0.181
*APOE*	rs429358	19:44,908,684	missense	c.388T>C (p.Cys130Arg)	C		
*APOE*	rs7412	19:44,908,822	missense	c.526C>T (p.Arg176Cys)	T		
*APOE* genotype						E2/E2	−0.9
						E2/E3	−0.4
						E2/E4	−0.2
						E3/E3	0.0
						E3/E4	0.1
						E4/E4	0.2
PRS from Wang et al., 2016 [131]							
*PCSK9*	rs11206510	1:55,030,366	intergenic	T>C	C	T	0.090
*CELSR2*	rs12740374	1:109,274,968	3-’UTR	c.*919G>T	T	G	0.230
*APOB*	rs515135	2:21,063,185	intergenic	T>C	T	C	0.160
*ABCG8*	rs6544713	2:43,846,742	intronic	c.322+431T>C	T	T	0.150
*HMGCR*	rs3846663	5:75,359,901	intronic	c.2458-84C>T	T	T	0.070
*TIMD4*	rs1501908	5:156,971,158	intergenic	C>G	G	C	0.070
*HNF1A*	rs2650000	12:120,951,159	intergenic	C>A	A	A	0.070
*LDLR*	rs6511720	19:11,091,630	intronic	c.67+2015G>T	T	G	0.260
*SUGP1 (NCAN)*	rs10401969	19:19,296,909	intronic	c.1243+80T>C	C	T	0.050
*MAFB*	rs6102059	20:40,600,144	intergenic	C>T	T	C	0.060

^1^ Positions are relative to genome assembly GRCh38.p2. ^2^ Nomenclature following the Human Genome Variation Society (HGVS). When a variant affects the coding region, cDNA (protein) changes are specified. ^3^ The minor allele (MA) considering the global population in 1000 Genomes. ^4^ Effect size on LDL-c due to a single copy of the risk allele, except for the *APOE* genotype, in mmol/L for the PRS from Talmud et al. (taken from Appendix A in [129]), and in SD units (b-coefficient) for the PRS from Wang et al. (taken from Supplementary Table II in [131]).

## Data Availability

Not applicable.

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
