# Peer review of "Genetic Determinants of Plasma Low-Density Lipoprotein Cholesterol Levels: Monogenicity, Polygenicity, and “Missing” Heritability"

_biomedicines, 2021, doi:10.3390/biomedicines9111728_

Round 1
Reviewer 1 Report
Genetic Determinants of Plasma Low-Density lipoprotein cholesterol levels: monogenicity, polygenicity and “missing” heritability. In the current study, Martin Campos analyzed by review the genetic determinations of plasma LDL cholesterol levels. Although the topic is not really novel, the review is well done, the article is well writer and the author have realized a conscientious bibliographic work. Although it is true that carrying out a systematic review of this field would have provided much more scientifically solid results. The author has carried out a good bibliographic search of the main articles in each area and field, summarizing novel articles in each field. Nevertheless, I have some minor’s comments: - Regarding of the prevalence of ADH in Spain, the article 38 (Zamora et al. J Clin Lipidology, 2017), defined ADH patients according to the levels of cholesterol adjusted by age. Maybe for this reason they found more prevalence than the estimate by other articles, like the estimate by Vallejo-Vaz (article 37) or by Lamiquiz-Moneo et al. Rev Esp Cardiol (Engl Ed) 2021, which estimate that prevalence of ADH is or 1:300 or 1:282, respectively in Spain population. In my opinion, the exclusive use of untreated LDL cholesterol level adjusted by age is not enough to classification of AHD patient. It is important to include other variables in the diagnosis of the ADH patient, such as the presence or not of previous CD in the family, the family history of hypercholesterolemia, the presence of xanthomas or corneal arch, etc. I consider that the author should comment this point in the article - Regarding to the effect of STAP1 as causal gene of ADH. Although the author cited a very good recent study in mice, I consider that he should cited too other articles developing with human, like: Blanco-Vaca et al. Clin Chim Acta 2018and Lamiquiz-Moneo et al. Atheroclerosis 2020. - Regarding to the effect of ABCG5/ABCG8 exist more recently articles which explained that part of patient with ADH fenotype have rare mutations in these genes, which produced them more absorption of cholesterol absorption, which could explain the origin of the dyslipidaemia in these patients (Lamiquiz-Moneo et al. 2017. J Clin Lipidology. ABCG5/G8 gene is associated with hypercholesterolemias without mutation in candidate genes and noncholesterol sterols; Laurens F Reeskamp. ABCG5 and ABCG8 genetic variants in familial hypercholesterolemia. J Clin Lipid. Mar-Apr 2020).Author Response
Review 1
Genetic Determinants of Plasma Low-Density lipoprotein cholesterol levels: monogenicity, polygenicity and “missing” heritability. In the current study, Martin Campos analyzed by review the genetic determinations of plasma LDL cholesterol levels. Although the topic is not really novel, the review is well done, the article is well writer and the author have realized a conscientious bibliographic work. Although it is true that carrying out a systematic review of this field would have provided much more scientifically solid results. The author has carried out a good bibliographic search of the main articles in each area and field, summarizing novel articles in each field. Nevertheless, I have some minor’s comments:
Thank you very much for the comments and suggestions.
- Regarding of the prevalence of ADH in Spain, the article 38 (Zamora et al. J Clin Lipidology, 2017), defined ADH patients according to the levels of cholesterol adjusted by age. Maybe for this reason they found more prevalence than the estimate by other articles, like the estimate by Vallejo-Vaz (article 37) or by Lamiquiz-Moneo et al. Rev Esp Cardiol (Engl Ed) 2021, which estimate that prevalence of ADH is or 1:300 or 1:282, respectively in Spain population. In my opinion, the exclusive use of untreated LDL cholesterol level adjusted by age is not enough to classification of AHD patient. It is important to include other variables in the diagnosis of the ADH patient, such as the presence or not of previous CD in the family, the family history of hypercholesterolemia, the presence of xanthomas or corneal arch, etc. I consider that the author should comment this point in the article.
Effectively, Zamora et al. 2017 considered as a threshold the validated age-adjusted LDL-c since the Information System for the Development of Research in Primary Care (SIDIAP) does not include enough information to apply the Dutch Lipid Clinic Network (DLCN) diagnostic score. Other studies considered genetic, clinical (mainly DLCN and Simon Broome), and others as the diagnostic criteria.
Not all the clinically diagnosed ADH patients presented a pathogenic mutation in a known gene, so it may be expected a lower prevalence applying a genetic than clinical diagnostic criteria. The meta-analysis of Beheshti et al. 2020 analyzed data from publication records in subgroups that followed different diagnostic criteria. They found a concordance between clinical and genetic diagnosis in the general population (figure 3 of their paper). However, in the ischemic heart disease (IHD), premature IHD, and hypercholesterolemic groups the genetic diagnosis showed lower prevalence than clinical diagnosis (figures 4, 5, and supplementary figure 5, respectively).
As noted by the reviewer, the different estimates of the ADH prevalence should be compared with caution because of the heterogeneity in the diagnostic criteria used. I have commented on this point in the manuscript.
- Regarding to the effect of STAP1 as causal gene of ADH. Although the author cited a very good recent study in mice, I consider that he should cited too other articles developing with human, like: Blanco-Vaca et al. Clin Chim Acta 2018and Lamiquiz-Moneo et al. Atheroclerosis 2020.
I briefly comment these two papers that reinforce the idea that STAP1 is unlikely a candidate gene.
- Regarding to the effect of ABCG5/ABCG8 exist more recently articles which explained that part of patient with ADH fenotype have rare mutations in these genes, which produced them more absorption of cholesterol absorption, which could explain the origin of the dyslipidaemia in these patients (Lamiquiz-Moneo et al. 2017. J Clin Lipidology. ABCG5/G8 gene is associated with hypercholesterolemias without mutation in candidate genes and noncholesterol sterols; Laurens F Reeskamp. ABCG5 and ABCG8 genetic variants in familial hypercholesterolemia. J Clin Lipid. Mar-Apr 2020).
I added a comment of the two references and added two more studies about ABCG5/8 in ADH.
Reviewer 2 Report
The author have done a great work in compiling the recent literature with respect to Plasma Low-Density Lipoprotein Cholesterol Levels. It will appreciated and the review would benefit from the following additions:
- The section cholesterol metabolism is already well understood. It is just re-capitulation of already known facts and not much work has been done in this section. Please quote recent literature and scientific developments in the field and how this field has shaped up around clinical relevance of low-density lipoprotein levels.
- The figure illustration is not appropriate. It is recommend to summarize the understanding authors have gained from literature and provide scientific illustration of the mechanistic understanding.
- Please provide a section with challenges in this field and what are the steps which can be taken with respect to future direction.
I would appreciate work from author's end to compile this work.
Author Response
The author have done a great work in compiling the recent literature with respect to Plasma Low-Density Lipoprotein Cholesterol Levels. It will appreciated and the review would benefit from the following additions:
Thank you very much for the reviewer comments. I tried to improve the manuscript following his suggestions.
- The section cholesterol metabolism is already well understood. It is just re-capitulation of already known facts and not much work has been done in this section. Please quote recent literature and scientific developments in the field and how this field has shaped up around clinical relevance of low-density lipoprotein levels.
I wrote the cholesterol metabolism section as only a brief overview of the endogenous and exogenous pathways as part of the introduction and a starting point for a reader unfamiliar with cholesterol metabolism. It is only an initial landscape. I based the review on genetic determinants of plasma LDL-c levels, and I have tried to structure it based on progressive advances in genetic analysis techniques. Enzymes, receptors, transcription factors, and other actors involved in cholesterol metabolism increase as new genes are identified.
I believe that an updated and extensive version of cholesterol metabolism at the beginning of the manuscript would break the structure of the following sections. I based the manuscript on the progressive advances that have been made in the genetics of plasma LDL-c and how they have expanded the initial landscape.
On the other hand, the manuscript does not pretend to be a systematic review of current knowledge about LDL-c metabolism. An extensive description of diseases associated with variations in plasma LDL-c and the clinical utility of nongenetic factors are out of the scope of this review. As it is a review of genetic factors, a possible clinical utility was discussed only when the findings are involved in hyper- or hypocholesterolemia genetic diagnosis.
I agree with the reviewer that a compilation of the contribution of genetic studies to the knowledge of LDL metabolism would improve the manuscript (see the next point).
- The figure illustration is not appropriate. It is recommend to summarize the understanding authors have gained from literature and provide scientific illustration of the mechanistic understanding.
Connecting with the previous comment, I have added a figure (Figure 1) summarizing the contribution of high-throughput studies in understanding the genetic factors associated with the plasma concentration of LDL-c. I have kept figure 1 (now Figure 2) because it shows in a very visual way the information that GWAS studies provide.
- Please provide a section with challenges in this field and what are the steps which can be taken with respect to future direction.
At the end of the manuscript, section 6 (The future: The era of system genetics) deals with future challenges, such as including gene-gene and gene-environment interactions in genetic studies and the development of systems genetics. I have expanded the section with a few more concepts.
I would appreciate work from author's end to compile this work.